# Strain-Tuned Spin-Wave Interference in Micro- and Nanoscale Magnonic Interferometers

**DOI:** 10.3390/nano12091520

**Published:** 2022-04-30

**Authors:** Andrey A. Grachev, Alexandr V. Sadovnikov, Sergey A. Nikitov

**Affiliations:** 1Saratov State University, 410012 Saratov, Russia; andrew.a.grachev@gmail.com (A.A.G.); nikitov@cplire.ru (S.A.N.); 2Kotel’nikov Institute of Radioengineering and Electronics, 125009 Moscow, Russia

**Keywords:** spin waves, magnonics, straintronics, Brillouin light scattering, Mach–Zehnder interferometer, spin-wave computing, yttrium-iron garnet

## Abstract

Here, we report on the experimental study of spin-wave propagation and interaction in the double-branched Mach–Zehnder interferometer (MZI) scheme. We show that the use of a piezoelectric plate (PP) with separated electrodes connected to each branch of the MZI leads to the tunable interference of the spin-wave signal at the output section. Using a finite element method, we carry out a physical investigation of the mechanisms of the impact of distributed deformations on the magnetic properties of YIG film. Micromagnetic simulations and finite-element modelling can explain the evolution of spin-wave interference patterns under strain induced via the application of an electric field to PP electrodes. We show how the multimode regime of spin-wave propagation is used in the interferometry scheme and how scaling to the nanometer size represents an important step towards a single-mode regime. Our findings provide a simple solution for the creation of tunable spin-wave interferometers for the magnonic logic paradigm.

## 1. Introduction

Recent advances in the fabrication of micro- and nanoscale magnetic structures based on insulating materials represent a promising alternative to the use of CMOS technologies for signal processing [1,2]. The development of spintronics and magnonics has led to the creation of micro- and nanostructures that are promising candidates for new components of magnetic memory and logic in devices with ultra-low energy consumption, especially in neuromorphic computing [3,4]. Micro- and nanoscale magnetic structures serve as a promising basis for the development of magnonic integrated circuits [5,6] compared to CMOS circuits and technologies [7,8,9]. The quasiparticles associated with the eigen excitations in magnetic materials that are known as spin waves (SWs) are magnons [3,4]. Magnons can be used as information carriers in magnonic circuits.

The phenomena of the constructive and destructive interference of SW in magnonic circuits can be used to fabricate logical devices [10,11,12,13,14,15]. This paradigm opens up the possibility to fabricate frequency-selective multiplexers and demultiplexers, as well as spin-wave directional couplers and splitters. Importantly, it has been demonstrated experimentally that coupling between SWs (or SW beams) can be used to manipulate magnon transport [16,17,18]. In such a concept, a Mach–Zehnder interferometer (MZI) [19,20,21,22,23,24] is a component of magnonic networks which can be used as a phase shifter.

The ability to manipulate the transfer of spin waves through electric and magnetic fields allows the development of more efficient magnonic devices [25,26,27,28,29,30,31,32]. This provision stimulates research in this area and also leads to the need to search for new information processing technologies, one of which should be straintronics. One area of interest relating to magnon straintronics [33,34,35] is the use of layered structures with two-dimensional deformations in the form of magnetostrictive and piezoelectric layers that are mechanically connected to each other. Recent theoretical and experimental studies show that deformation can be used to create energy-efficient complex two-dimensional and three-dimensional piezoelectric materials and heterostructures based on semiconductors [36,37], ferroelectrics [38,39,40], piezoelectric crystals, and ceramics [33,41,42]. It has also been shown that it is possible to induce a ferromagnetic resonance frequency shift due to the influence of the conversion of an electric field into a magnetic field [41,43]. The influence of the electric field on the magnetic configuration occurs due to the modification of the effective internal magnetic field. The latter changes due to reverse magnetostriction (Villari effect) as a result of the local deformation of the magnetic film.

Here, we report the experimental study of spin-wave propagation and interaction in a double-branched MZI scheme. We show that the use of PP with separated electrodes connected to each branch of MZI leads to the tunable interference of the spin-wave signal at the output section. Using numerical and experimental techniques, we show the electric field tunability of the spatial and transfer characteristics of dipole spin waves in considered heterostructures. Using a finite element method, we conduct a physical investigation of the mechanisms of the impact of distributed deformations on the magnetic properties of YIG film. Using micromanetic calculations, we carry out a comparison between microsized and nanosized configurations of MZI.

## 2. Structure Fabrication

A sketch of the investigated structure is shown in Figure 1a. The considered structure represents a irregular ferrite stripe in a Mach–Zehnder interferometer (MZI) configuration with the width w=500μm. The interferometer was fabricated from 10μm-thick monocrystalline ferrite YIG film with a saturation magnetization of $M_{s}=139$ G using the laser scribing technique [44,45]. The interferometer was located on the 500μm-thick gallium gadolinium garnet (GGG) substrate. The interferometer branches were formed by etching an oval hole in the center of YIG film. The length of MZI was $L_{1}=6$ mm. We used a 200μm-thick piezoelectric layer of lead zirconate titanate (PZT) as a mechanism for creating elastic deformations. A 1μm-thick titanium electrode was placed (“GND” in Figure 1a) on the top side of PZT, which did not have a significant effect on the propagation of SW in MZI. On the other side of the PZT, two trapezoidal 100 nm-thick titanium electrodes were formed in the areas under the top and bottom branches of MZI, and a PZT layer was bonded to a YIG film using an OMEGA TT300 cement heat-cured, two-part epoxy adhesive based on ethyl-cyanoacrylate $C_{6}H_{7}NO_{2}$ for strain measurement. A voltage of $V_{c}$ was applied to the electrodes in the experiment. For the more efficient control of spin-wave transport using local deformations, we used a spatial resolution laser ablation technique to etch a piezoelectric layer that was 25μm thick. The SW was excited using a microstrip antenna that was 1μm thick and 30μm wide. The structure was placed in an external static magnetic field, $H_{0}=1490$ Oe, oriented along the *x* axis to effectively excite the magnetostatic surface wave (MSSW) in MZI.

## 3. Numerical Calculations

To conduct a physical investigation of the mechanisms of the impact of distributed deformations on the magnetic properties of YIG film, we developed a numerical model based on the finite element method (FEM) [46] in order to calculate the mechanical deformations created by the PZT layer, taking into account the transformation of the internal magnetic field in MZI. The obtained transformations of the internal magnetic field were taken into account in micromagnetic calculations [47]. At the stage of solving the magnetostrictive problem [48], it was assumed that the magnetostrictive effect could be modeled using the nonlinear dependence of magnetostriction on the magnetization **M** and mechanical stress **S** in the material. The stress in YIG is calculated using:(1)$$S=c_{H}\left[\varepsilon_{el}-\varepsilon_{me}\left(M\right)\right],$$
where the stiffness matrix $c_{H}$ is determined by two parameters: Young’s modulus $(E=2\times 10^{12}$ Pa) and Poisson’s ratio (ν=0.29). The relation for magnetostrictive deformation $\varepsilon_{me}$ is represented as a quadratic isotropic magnetization function **M**:(2)$$\varepsilon_{me}=\frac{3}{2}\frac{\lambda_{S}}{M_{s}^{2}}\mathrm{dev}(M⊗M),$$
where the tensor product of two vectors is defined as ${\left(M⊗M\right)}_{ij}=M_{i}M_{j}$, and $\lambda_{S}=-2.2\times 10^{-6}$ is the saturation magnetostriction, which is the maximum magnetostrictive deformation achieved with saturation magnetization $M_{s}$. Nonlinear magnetization in the magnetostrictive material is found from the following nonlinear relation:(3)$$M=M_{S}L\left(\left|H_{\mathrm{eff}}\right|\right)\frac{H_{\mathrm{eff}}}{\left|H_{\mathrm{eff}}\right|}$$
where *L* is the Langevin function andthe effective magnetic field in the material is given by:(4)$$H_{\mathrm{eff}}=H_{0}+\frac{3\lambda_{S}}{\mu_{0}M_{S}^{2}}S_{ed}M$$
where $\mu_{0}$ is the magnetic permeability of free space and the deviatoric elastic stress tensor is related to the elastic strain $\varepsilon_{el}$ in the material through $S_{\mathrm{ed}}=\mathrm{dev}\left(C_{H}\varepsilon_{el}\right)$. The second term in (4) describes the mechanical stress contribution to the $H_{\mathrm{eff}}$, which is associated with the Villari effect.

The computational domain was divided into finite elements of tetrahedral form with the smallest element side size of 25 nm in the region of the electrodes. The relative variation in the size of the PZT layer is shown in inset of Figure 1a, where the color gradient shows the distribution of the components of the mechanical stress tensor $S_{xx}$ in the case of $E_{1}=10$ kV/cm and $E_{2}=-10$ kV/cm. It follows that the deformation of the piezoelectric layer occurs in the local region of the PZT layer under the electrodes, leading to a change in the value of the internal magnetic field $H_{int}$ in the MZI due to the inverse magnetostrictive effect. Figure 1b shows the profile $H_{int}=\left|H_{int}\left(x\right)\right|$ in the case of zero, positive, and negative electric field values, which applies to the different branches of MZI. The lateral confinement of magnonic stripe in the *x*-direction leads to a nonuniform distribution of the internal magnetic field. In this case, the influence of elastic deformations leads to the transformation of the internal magnetic field. The dashed lines in Figure 1b show the $H_{int}$-profiles when a positive field is applied to the top branch and a negative field to the bottom branch. It should be noted that a significant contribution of elastic deformation is represented by changes along the *z* and *x* coordinates. In this case, a value in the order of 3 Oe is added to the *z*-component of the internal magnetic field. The resulting addition to $H_{int}$ is 7 Oe. Thus, using elastic deformations on the YIG film leads to the transformation of the internal magnetic field, followed by transformation in the spin-wave dispersion in the interferometer branches.

Next, using a micromagnetic calculations based on the solution of the Landau–Lifshitz–Gilbert equation [47,49], we investigated a voltage-controlled transformation of spin-wave dynamics and interference. The parameters of the YIG were: saturation magnetization $M_{s}=139$ G, exchange constant $A_{ex}=3.612$ pJ/m, and Gilbert damping $\alpha =5\times 10^{-5}$. The damping at the ends of the simulated structure and the high damping absorber was set to exponentially increase to 0.5 to prevent spin-wave reflection. The static magnetization oriented itself perpendicular to the MZI spontaneously to excite an MSSW. The mesh was set to 5×3×2 μm${}^{3}$. To excite propagating spin waves, a sinusoidal magnetic field $h=h_{0}sin\left(2\pi ft\right)$ was applied over an area of 100 μm in length, with a varying oscillation amplitude $h_{0}$ and microwave frequency *f*. The $M_{z}\left(x,y,t\right)$ of each cell was collected over a period of 300 ns, which is long enough to reach the steady state. The fluctuations $m_{z}\left(x,y,t\right)$ were calculated for all cells via $m_{z}\left(x,y,t\right)=M_{z}\left(x,y,t\right)-M_{z}\left(x,y,0\right)$, where $M_{z}\left(x,y,0\right)$ corresponds to the ground state. The influence of the external electric field was taken into account by creating regions with a reduced (increased) value of the internal magnetic field in the regions of the interferometer branches. The left column of Figure 2 shows the SW intensity distribution $I\left(x,y\right)=m_{x}^{2}+m_{z}^{2}$ in the case of MSSW excitation at f=6.23 GHz. It should be noted that such a configuration of magnonic structure allows us to realize a NOT logic function. SWs generated from the one end are allowed to be split into the top and bottom branches and then merged to interfere on the other end, without or by applying an electric field to the PZT layer. In the case of zero electric fields $E_{1}=E_{2}=0$ kVcm, representing the logical input, the two merged SW beams constructively interfere with each other due to the lack of differences in phase (see Figure 2c,d) and amplitude (see Figure 2a,b) between the two, representing the “1” logical output seen at the top of Figure 2a. By contrast, with electric fields of $E_{1}=10$ kV/cm and $E_{2}=10$ kV/cm, the split SWs are merged to destructively interfere on the right end because they, after passing the upper and lower branches, become out of phase with each other, signifying the 0 logical output in Figure 2b. To demonstrate an efficiently voltage-controlled variation in spin-wave amplitude and phase, we plot an integrated value of dynamic magnetization $m_{z}$-component in the output section (in the cross section x=4 mm) as a function of the external electric field applied to the PZT layer at the fixed frequency f=6.23 GHz. As a result, the phase shift ΔΦ(E,f), created during the separate SW propagation between the top and bottom branches, is given by $\Delta \Phi \left(E,f\right)=\int k_{top}\left(l\right)-k_{bot}\left(l\right)dl$, where $k_{top}\left(l\right)$ and $k_{bot}\left(l\right)$ are the SW dispersions in the top and bottom branches of MZI. The ΔΦ(E,f) value can be controlled by varying *E* and is also variable for different *f*, as seen in Figure 2e. Applying a positive or negative electric field leads to a decrease in the magnitude of the phase and amplitude in the output section of the MZI in a periodic manner. Thus, we can obtain a voltage-controlled switching of spin-wave amplitude and phase in the proposed MZI.

At the same time, nano-scale single-mode magnonic waveguides can also overcome the issue of parasitic magnon scattering into higher modes. Furthermore, reducing the dimensions of magnonic structures to the atomic scale could potentially shift the frequency of the spin waves from the GHz to the THz range. Next, let us consider a MZI in at a sub-micron size. To simulate the processes of the propagation of spin-wave excitations in a nano-scale MZI, the following dimensions were chosen: the width w=2μm, thickness t = 40 nm, and length $L_{1}=20\;\mu$m. The width of the microwave signal excitation source was 500 nm. A sinc-field pulse $h_{y}=h_{0}sinc\left(2\pi f_{c}t\right)$, with an oscillation field of $h_{0}=0.1$ Oe and a cutoff frequency of $f_{c}=10$ GHz, was used to excite a wide range of spin waves. To compare the micro- and nano-scaled MZIs, we plotted the frequency-dependent spectral power densities for a previously considered micro-scaled MZI (see the red solid curve in Figure 3a) and nano-scaled MZI (see the red solid curve in Figure 3b). A decrease in geometric dimensions leads to an increase in the effect of shape anisotropy for nano-scale waveguides. This leads to the transformation of the transmitted spin waves through the MZI. Applying an external electric field (see the blue dashed curves in Figure 3) for a nano-scale MZI leads to a variation of the frequency dips in the spin-wave transmission, which correspond to destructive interference at the output.

## 4. Experimental Observation of Voltage-Controlled Manipulation of Spin-Wave Amplitude

The experimental study of the dynamics of the MSSW was carried out using the Brillouin light scattering spectroscopy (BLS) of magnetic materials in the backscattering configuration, as shown in our previous works [33,39,50,51]. The BLS experiments were performed using a Sandercock (3+3)-pass tandem Fabry–Perot interferometer. A beam of 50 mW of P-polarized green light from a single-mode solid state laser Excelsior (EXLSR-532-50-CDRH) was focused on the sample surface using a Nikkor objective with a numerical aperture of 1.2 and focal length of 50 mm. The scattered light was collected using the same objective. The technique of signal suppression from elastically scattered and surface phonon-scattered light was used to discriminate the contribution of magnons to the signal from elastically scattered light. Spatial maps of spin-wave intensity $I_{BLS}$ were obtained by scanning with the probing light spot along the *x* and *y* directions with a resolution of 25μm and then integrating them over the pulse repetition period. SW in the structure were excited using a microstrip antenna with a thickness of 1μm (in the *z*-direction ) and a width of 30μm (in the *y*-direction). In order to excite SW, we applied a microwave current to the input antenna. The current was in the form of a train of 100 ns-long microwave pulses with repetition period of 2 μs. We chose this pulse regime in order to avoid the heating of MZI in the vicinity of the shortened microwave antenna. The intensity of the scattered light scales was the SW intensity. The BLS intensity maps are shown for different values of external electric fields in Figure 4 for a fixed frequency f=6.23 GHz. The application of an external electric field changes the magnitude of the internal magnetic field in the MZI branches, thereby changing the SW dispersion in each of the branches. In this case, one can also notice a change in the wavelength in the branches and a change in the nature of the spin-wave interference in the output section of the MZI. The idea of controlling the modes of spin-wave interference, obtained by the method of micromagnetic modeling, was confirmed experimentally using Brillouin light scattering spectroscopy.

This concept of the intensity and phase signal manipulation indicates the possibility of using the proposed MZI device as a basic element of signal processing systems based on the principles of neuromorphic [18,52,53,54,55,56] and magnonic logic, such as: magnon logic cells based on fuzzy logic elements, neuromorphic multiplexing and demultiplexing systems [57], and space–frequency dividers and couplers [1] of information signals in the microwave and terahertz wavelength range.

## 5. Conclusions

Using numerical and experimental techniques, we investigated the propagation and interaction of SWs propagating as guided modes of YIG stripes in a double-branched MZI scheme. We showed that the use of the PP with separated electrodes connected to each branch of MZI led to the tunable interference of the spin-wave signal at the output section. Micromagnetic simulations and finite-element modelling explained the evolution of the spin-wave interference pattern under the strain induced via the application of an electric field to the PP electrodes. We showed how the multimode regime of spin-wave propagation was used in the interferometry scheme and how the scaling to the nanometer size represents an important step towards the creation of a single-mode regime. Using numerical methods, a comparison was made between the microsize and nanosize configurations of MZI. It was shown that a change in the size of the MZI leads to a transformation of the SW propagation patterns, including a change in the nature of the SW interference. Our findings provide a simple solution for tunable spin-wave interferometers for the magnonic logic paradigm.

## Figures and Tables

**Figure 1 nanomaterials-12-01520-f001:**
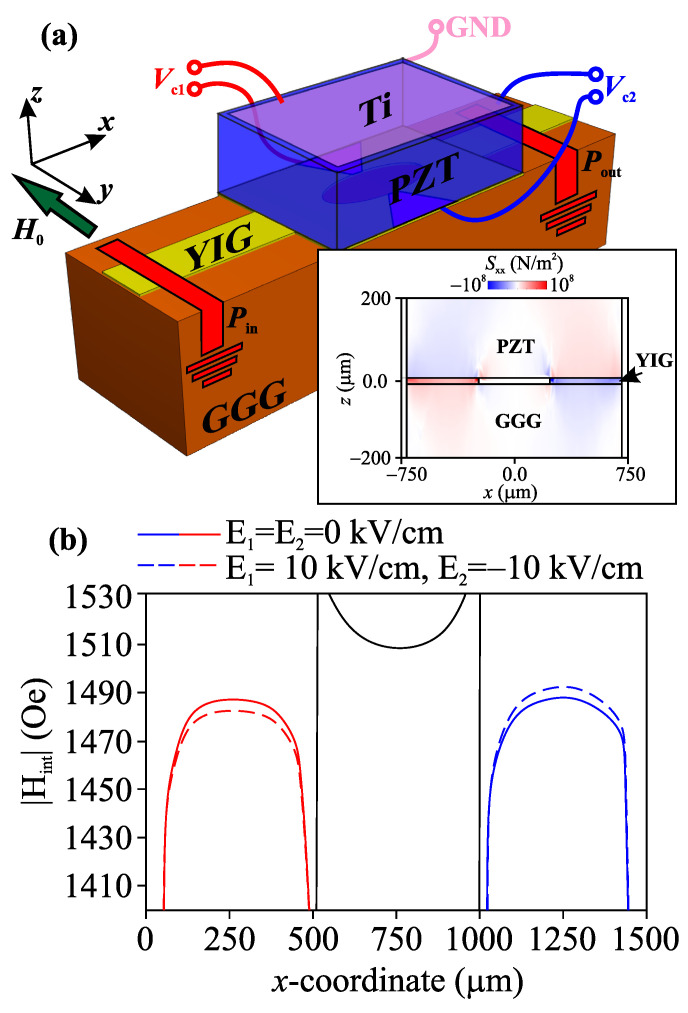
(**a**) Scheme of the considered MZI. Inset: The distribution of the mechanical stress tensor component $S_{xx}$ when $E_{1,2}=\pm 10$ kV/cm was applied to the electrodes. (**b**) Internal magnetic field profiles $H_{\mathrm{int}}$ where $E_{1,2}=0$ kV/cm (solid curves) and $E_{1,2}=\pm 10$ kV/cm (dashed curves) at $H_{0}=1100$ Oe. The black lines on the graph represent the distribution of the magnetic field at the boundary of the YIG stripe.

**Figure 2 nanomaterials-12-01520-f002:**
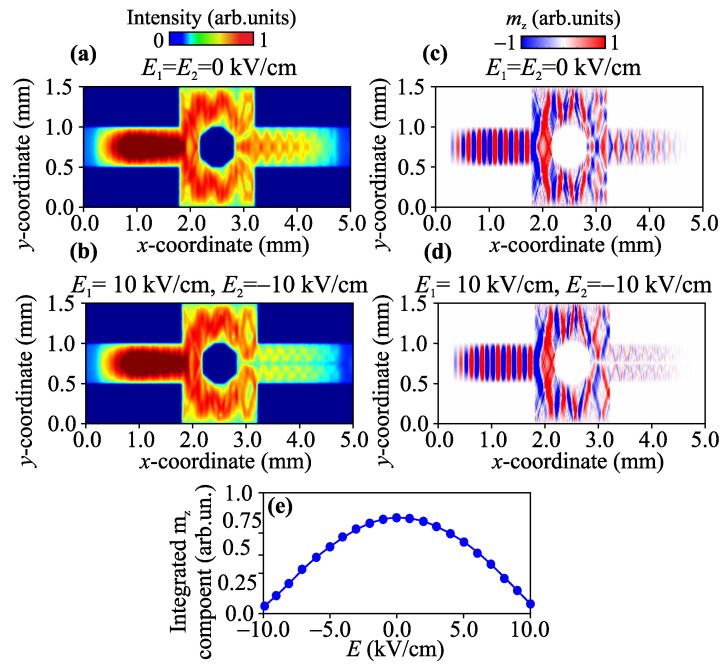
Spin-wave intensity maps (**a**,**b**) and $m_{z}$-component (**c**,**d**) of the dynamic magnetization for $E_{1,2}=0$ kV/cm (**a**,**c**) and $E_{1,2}=\pm 10$ kV/cm (**b**,**d**). (**e**) Integrated value of the dynamic magnetization $m_{z}$-component in the output section as a function of the external electric field.

**Figure 3 nanomaterials-12-01520-f003:**
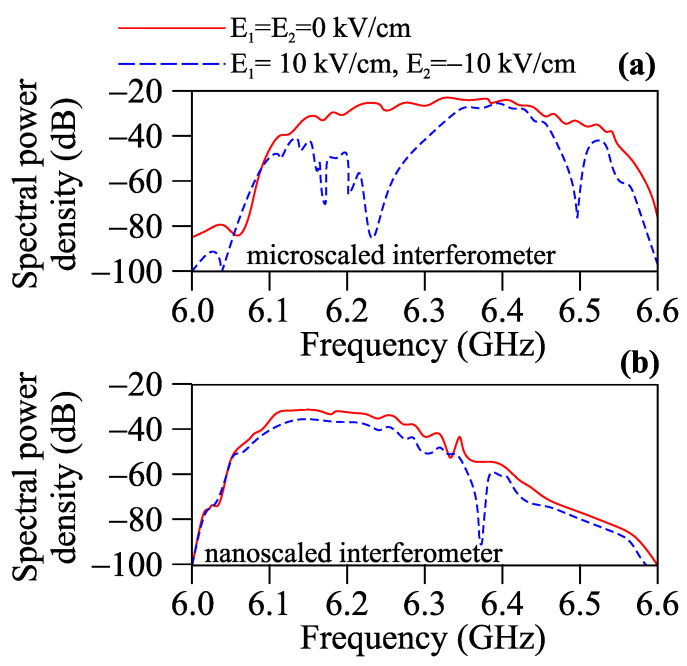
Frequency dependencies of spectral power density for micro-scaled (**a**) and nano-scaled (**b**) MZI in the case of different values of *E*.

**Figure 4 nanomaterials-12-01520-f004:**
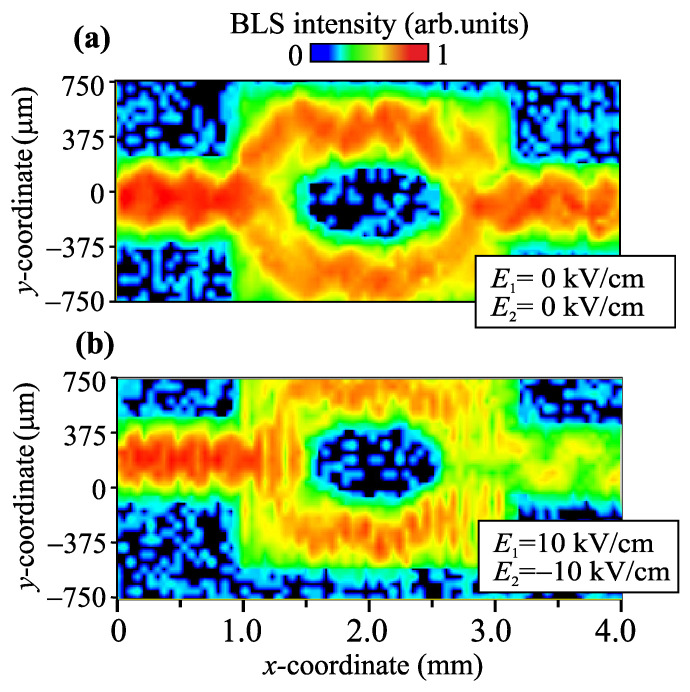
2D spatial maps of BLS intensity $I_{BLS}$ in the case of $E_{1,2}=0$ kV/cm (**a**) and $E_{1,2}=\pm 10$ kV/cm (**b**).

## Data Availability

The data that support the results of this study are available from the corresponding author upon reasonable request.

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
