# Peer review of "Strain-Tuned Spin-Wave Interference in Micro- and Nanoscale Magnonic Interferometers"

_nanomaterials, 2022, doi:10.3390/nano12091520_

Round 1
Reviewer 1 Report
The manuscript “Strain-tuned spin-wave interference in micro- and nanoscale magnoninc interferometers” by A. A. Grachev et al reported the the numerical simulation and experimental study of spin-wave propagation and interaction in the double-branched Mach-Zehnder interferometer scheme. The content is interesting, the manuscript could be published after addressing the following issues:
- Experimental method with more details should be provided.
- Figure 1, 2, 4 should be checked and revised.
- More experimental study and discussion should be provided.
Author Response
04 April 2022
“Strain-tuned spin-wave interference in micro- and nanoscale magnonic interferometers”
Dear Editor,
Dear Editorial Office Staff,
We thank the Referee for the high appraisal of our work, and for the recommendation that our work can be published in “Nanomaterials”.
Reviewer #1
The manuscript “Strain-tuned spin-wave interference in micro- and nanoscale magnonic interferometers” by A. A. Grachev et al reported the numerical simulation and experimental study of spin-wave propagation and interaction in the double-branched Mach-Zehnder interferometer scheme. The content is interesting, the manuscript could be published after addressing the following issues:
- Experimental method with more details should be provided.
Reply: We thanks Referee for the question, we added more detailed description of experimental method. In particular, this refers to a more detailed description of SW excitation in a YIG film.
SW in the structure were excited using a microstrip antenna with a thickness of $1~\mu m$ (in the $z$-direction ) and a width of $30~\mu m$ (in the $y$-direction). In order to excite SW, we applied a microwave current to the input antenna. The current was in the form of a train of 100-ns-long microwave pulses with repetition period of 2~$\mu$s. We chose the pulse regime in order to avoid heating of MZI in the vicinity of the shortened microwave antenna.
- Figure 1, 2, 4 should be checked and revised.
Reply: Thanks for the remark. We have changed Figure 1, and also changed the captions for Figures 2 and 4.
- More experimental study and discussion should be provided.
Reply: Thanks for the remark. We have added to Section 4 an additional description of the experimental design, as well as an additional discussion regarding the comparison of numerical and experimental results.
In order to excite SW, we applied a microwave current to the input antenna. The current was in the form of a train of 100-ns-long microwave pulses with repetition period of 2~$\mu$s. We chose the pulse regime in order to avoid heating of MZI in the vicinity of the shortened microwave antenna. The intensity of the scattered light scales as the SW intensity. The BLS intensity maps are shown for different values of external electric fields in Fig. 4 for a fixed frequency $f = 6.23$~GHz. The application of an external electric field changes the magnitude of the internal magnetic field in the MZI arms, thereby changing the SW dispersion in each of the arms. In this case, one can also notice a change in the wavelength in the arms and a change in the nature of the spin-wave interference in the output section of the MZI. The idea of controlling the modes of spin-wave interference, obtained by the method of micromagnetic modeling, is confirmed experimentally with Brillouin light scattering spectroscopy.
This concept of the intensity and phase signal manipulation indicates the possibility of using the proposed MZI device as a basic element of signal processing systems based on the principles of neuromorphic~\cite{Torrejon2017,Markovi2019,Romera2018,Chen2019,Yu2020,Csaba2017} magnonic logic, such as: magnon logic cells based on fuzzy logic elements, neuromorphic multiplexing and demultiplexing systems~\cite{Bracher18}, space-frequency dividers and couplers~\cite{Wang2020} of information signals in the microwave and terahertz wavelength range.
However, we did not add additional experimental results due to the short response time.
We thank you in advance for your kind consideration.
On behalf of all authors,
Andrey Grachev

Reviewer 2 Report
The paper by AA Grachev et al reports on a simple example of spin-wave interferometer, the spin-wave can be splitted at the output of a magnonic logic interferometer by simply control of the strain of PZT layer.
The result sounds convincing but authors are not careful in preparing their manuscript. Therefore, I am not recommending for publication in the present form.
Several points authors should consider in their revision:
- Carefully check the title, there is an error in spelling
- Authors should summary their achievement in the abstract in a quantitative way
- Motivation should be direct and clear
- The link between simulation and experimental data should be clear
- Check grammatical errors in the whole MS.
Author Response
04 April 2022
“Strain-tuned spin-wave interference in micro- and nanoscale magnonic interferometers”
Dear Editor,
Dear Editorial Office Staff,
We thank the Referee for the high appraisal of our work, and for the recommendation that our work can be published in “Nanomaterials”.
Reviewer #2
The paper by AA Grachev et al reports on a simple example of spin-wave interferometer, the spin-wave can be splitted at the output of a magnonic logic interferometer by simply control of the strain of PZT layer. The result sounds convincing but authors are not careful in preparing their manuscript. Therefore, I am not recommending for publication in the present form.
Several points authors should consider in their revision:
- Carefully check the title, there is an error in spelling
Reply: We thanks Referee for the remark. We've corrected the typo in the title.
«Strain-tuned spin-wave interference in micro- and nanoscale magnonic interferometers»
- Authors should summary their achievement in the abstract in a quantitative way
Reply: Thanks for the question. We supplemented the abstract with the obtained results.
Here we report on the experimental study of spin-wave propagation and interaction in the double-branched Mach-Zehnder interferometer (MZI) scheme. We show that the use of the piezoelectric plate (PP) with the separated electrodes connected to each branch of MZI leads to the tunable interference of the spin-wave signal at the output section. Using a finite element method we made a physical a physical interpretation the mechanisms of the impact of distributed deformations on the magnetic properties in YIG film. Micromagnetic simulations and finite-element modelling explain the evolution of spin-wave interference pattern under the strain induced via the electric field application to the PP electrodes. We show how the multimode regime of spin-wave propagation is used in the interferometry scheme and how the scaling to the nanometer sizes represents an important step towards the single mode regime. Our findings provide a simple solution for tunable spin-wave interferometers for the magnonic logic paradigm.
- Motivation should be direct and clear
Reply: Thanks for the remark. We have modified the introduction to include more current works on magnonics and straintronics. And we added a discussion in the final section regarding the application of the considered structures.
This concept of the intensity and phase signal manipulation indicates the possibility of using the proposed MZI device as a basic element of signal processing systems based on the principles of neuromorphic~\cite{Torrejon2017,Markovi2019,Romera2018,Chen2019,Yu2020,Csaba2017} magnonic logic, such as: magnon logic cells based on fuzzy logic elements, neuromorphic multiplexing and demultiplexing systems~\cite{Bracher18}, space-frequency dividers and couplers~\cite{Wang2020} of information signals in the microwave and terahertz wavelength range.
- The link between simulation and experimental data should be clear
Reply: Thanks for the remark. We have added to Section 4 an additional description of the experimental design, as well as an additional discussion regarding the comparison of numerical and experimental results.
In order to excite SW, we applied a microwave current to the input antenna. The current was in the form of a train of 100-ns-long microwave pulses with repetition period of 2~$\mu$s. We chose the pulse regime in order to avoid heating of MZI in the vicinity of the shortened microwave antenna. The intensity of the scattered light scales as the SW intensity. The BLS intensity maps are shown for different values of external electric fields in Fig. 4 for a fixed frequency $f = 6.23$~GHz. The application of an external electric field changes the magnitude of the internal magnetic field in the MZI arms, thereby changing the SW dispersion in each of the arms. In this case, one can also notice a change in the wavelength in the arms and a change in the nature of the spin-wave interference in the output section of the MZI. The idea of controlling the modes of spin-wave interference, obtained by the method of micromagnetic modeling, is confirmed experimentally with Brillouin light scattering spectroscopy.
- Check grammatical errors in the whole MS.
Reply: Thanks for the remark. We have tried to rework the text, removing most of the grammatical errors.
We thank you in advance for your kind consideration.
On behalf of all authors,
Andrey Grachev

Reviewer 3 Report
authors reported spin-wave propagation and interaction in a double-branched Mach-Zehnder interferometer scheme. They use PZT to tune the intereference of the spin wave. The observed spien wave signal is in reasonable agreement with the numerical simulations. The results are interesting and deserve publication.
Author Response
04 April 2022
“Strain-tuned spin-wave interference in micro- and nanoscale magnonic interferometers”
Dear Editor,
Dear Editorial Office Staff,
We thank the Referee for the high appraisal of our work, and for the recommendation that our work can be published in “Nanomaterials”.
We thank you in advance for your kind consideration.
On behalf of all authors,
Andrey Grachev

Round 2
Reviewer 1 Report
The revised manuscript can be accepted in Nanomaterials.
Author Response
20 April 2022
“Strain-tuned spin-wave interference in micro- and nanoscale magnonic interferometers”
Dear Editor,
Dear Editorial Office Staff,
We thank the Referee for the high appraisal of our work, and for the recommendation that our work can be published in “Nanomaterials”.
We have corrected grammatical and semantic inaccuracies. And also corrected the abstract and discussions in the text of the manuscript.
Reviewer 2 Report
The data presented in the paper is interesting, however, the authors should rework it for better readability. I found many grammatical errors and a lot of complicated sentences in the manuscript. Just a few examples: "To formulation a physical interpretation..."; "Using a finite element method we made a physical a physical interpretation the mechanisms of...."; "Based on the numerical simulation, a physical interpretation of the physical phenomenon of the transformation of the spectrum of eigenmodes of strain-tuned MZI will be given."
Also, you should avoid the use of the same sentence in the abstract and conclusion.
You should consider editing your manuscript with an English-speaking expert.
I require a major revision to check the revision once more before giving my final decision.
Author Response
20 April 2022
“Strain-tuned spin-wave interference in micro- and nanoscale magnonic interferometers”
Dear Editor,
Dear Editorial Office Staff,
We thank the Referee for the high appraisal of our work.
We have corrected grammatical and semantic inaccuracies. And also corrected the abstract and discussions in the revised version of the manuscript.
Round 3
Reviewer 2 Report
The manuscript is easy to read now, I recommend this revised MS for publication.